**RESEARCH**

# Eukaryotic-driven directed evolution of Cas9 nucleases

Giulia Vittoria Ruta[1*], Matteo Ciciani[1,2], Eyemen Kheir[1], Michele Domenico Gentile[1], Simone Amistadi[1,4], Antonio Casini[3] and Anna Cereseto[1*]

*Correspondence:
giuliavittoria.ruta@unitn.it; anna.
cereseto@unitn.it

[1] Laboratory of Molecular Virology, Department CIBIO, University of Trento, Trento, Italy
[2] Laboratory of Computational Metagenomics, Department CIBIO, University of Trento, Trento, Italy
[3] Alia Therapeutics S.R.L, Trento, Italy
[4] Present address: Laboratory of Chromatin and Gene Regulation During Development, Université de Paris, Imagine Institute, INSERM UMR 1163, Paris, France

## Abstract

**Background:** Further advancement of genome editing highly depends on the development of tools with higher compatibility with eukaryotes. A multitude of described Cas9s have great potential but require optimization for genome editing purposes. Among these, the Cas9 from *Campylobacter jejuni*, CjCas9, has a favorable small size, facilitating delivery in mammalian cells. Nonetheless, its full exploitation is limited by its poor editing activity.

**Results:** Here, we develop a Eukaryotic Platform to Improve Cas Activity (EPICA) to steer weakly active Cas9 nucleases into highly active enzymes by directed evolution. The EPICA platform is obtained by coupling Cas nuclease activity with yeast auxotrophic selection followed by mammalian cell selection through a sensitive reporter system. EPICA is validated with CjCas9, generating an enhanced variant, UltraCjCas9, following directed evolution rounds. UltraCjCas9 is up to 12-fold more active in mammalian endogenous genomic loci, while preserving high genome-wide specificity.

**Conclusions:** We report a eukaryotic pipeline allowing enhancement of Cas9 systems, setting the ground to unlock the multitude of RNA-guided nucleases existing in nature.

## Background

The development of CRISPR-Cas tools [1, 2] and the emerging number of RNA-guided prokaryotic and eukaryotic nucleases [3–5] are giving a tremendous impulse to the development of advanced therapies based on genomic modifications [6]. Yet, to further accelerate and broaden the application of genome editing [7], a larger set of tools is highly needed to match the complexity of gene therapy approaches. In particular, there is a demand to increase the accessibility to any genomic site by enlarging the repertoire of enzymes covering the largest number of protospacer adjacent motifs (PAM) [8, 9]. Moreover, the efficacy of cellular and organ delivery, which highly depends on the cargo size [10], is very relevant to the field, thus reinforcing the need for low molecular weight Cas enzymes. Small molecular size is indeed necessary for Cas compatibility with an adeno-associated viral vector (AAV) without enzyme splitting [11] and allows

the generation of compact base editors and prime editors, which can precisely edit DNA avoiding double-strand breaks [1, 2].

A large reservoir of CRISPR-Cas systems and novel emerging ancestral RNA-guided nucleases have been identified in metagenomic data banks. Nonetheless, the majority of enzymes isolated from bacteria [12], archaea [13], and viruses [14] are poorly active in mammalian cells. Strategies aimed at enhancing these systems for genome editing include molecular engineering of the nucleases by rational design [15], bacterial screening [16], or optimization of the sgRNA [17]. Despite these efforts, few CRISPR-Cas tools are sufficiently active for efficient genome editing in eukaryotes [18], as demonstrated by the paucity of RNA-guided nucleases for experimental use, as opposed to the multitude of systems identified in metagenomic data banks. To unlock the natural reservoir of prokaryotic systems, experimental setups are needed to enable the adaptation of enzymes to the eukaryotic nuclear environment. We have previously demonstrated that yeast cells are ductile eukaryotic systems that can improve Cas precision by directed evolution [19]. On this groundwork, we generated a Eukaryotic Platform to Improve Cas Activity (EPICA). To demonstrate the efficacy of this evolution platform, we introduced EPICA CjCas9 [20], a small size (984 aa) Cas enzyme with favorable features for genome editing, which is however minimally used due to its reduced nuclease activity [21]. The platform produced an enhanced variant, UltraCjCas9, carrying modifications hardly predictable through a rational engineering approach.

## Results

### Yeast-based directed evolution enhances Cas9 activity

To generate a eukaryotic platform for the evolution of Cas9 nuclease activity, a yeast strain was engineered by integrating two different cleavage reporter cassettes in the tryptophan (TRP) and adenine (ADE) genes. Disruption of the TRP and ADE genes by cassette insertion turns yeast growth dependent from adenine and tryptophan culture integration. The cassettes contain target sites including different PAM sequences compatible with the Cas9 under evolution, to allow the isolation of variants evolved independently from specific PAM sequence recognition. Homology arms were inserted at the edges of the cassettes to mediate single-strand annealing (SSA) repair induced by Cas9 cleavage of the target locus leading to gene reconstitution [19]. This model, schematized in Fig. 1a, allows the selection of active Cas9 molecules by auxotrophy through limiting growth factors conditions (TRP and ADE deprivation, TRP − and ADE − , respectively), permitting exclusive proliferation of cells with restored TRP and ADE loci generated by efficient Cas9 cleavage. The platform was initially tested with SpCas9, showing high levels of yeast cell survival in single ADE − and TRP − or combined ADE − /TRP − conditions; by contrast, wild-type CjCas9 (CjCas9 WT) showed 20% of living colonies in ADE − conditions but nearly absent survival in ADE/TRP deprived culture conditions (Fig. 1b). Transformation of a cDNA library of CjCas9 variants obtained through random PCR mutagenesis led to isolation of colonies growing in ADE − /TRP − conditions (Fig. 1c). Starting from the colonies obtained by transformation of the mutagenized CjCas9 library, the process of random mutagenesis and transformation in ADE − / TRP − conditions was reiterated until a decreasing trend in cell survival was observed (rounds 5 and 6), likely resulting from saturating mutagenesis. The cDNA isolated from

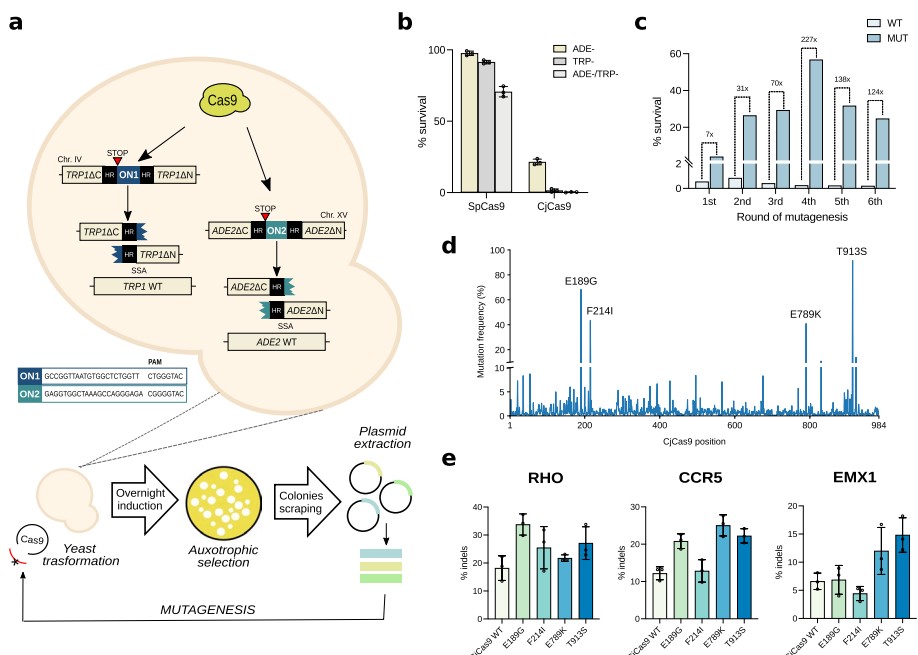

**Fig. 1** Directed evolution step to enhance Cas nuclease activity. **a** Schematic and experimental flowchart of the yeast Cas evolution step. Framed sequences correspond to the ON1 and ON2 cassettes carrying the CjCas9 target sites and AAVS1-TS34 and AAVS1-TS32 targets, respectively. Active mutants cleaving both ON1 and ON2 in the ADE2 and TRP1 loci are amplified from colonies growing in ADE − and TRP − conditions and undergo additional rounds of mutagenesis and selection. **b** Cleavage activity of SpCas9 and CjCas9 are reported as percentages of survival, corresponding to the number of colonies growing in TRP − and ADE − plates with respect to total transformants measured in unrestricted growth conditions. Culture plates shown in Additional file 1: Fig. S1a. **c** The activity of the CjCas9 variants (MUT) isolated at each round of evolution was compared with CjCas9 WT by measuring the percentages of survival after transformation. Culture plates are shown in Additional file 1: Fig. S1b. **d** Peaks at each CjCas9 amino acid position represent the frequency of the mutated residues obtained by NGS deep sequencing. Amino acid substitutions with high frequency are reported in the graph. **e** Editing activity at the endogenous loci in HEK293 cells of CjCas9 variants containing substitutions corresponding to the most frequently mutated amino acids in **d**. In **b** and **e**, data are reported as mean ± standard deviation of $n \geq 3$ biologically independent samples. Individual values are represented as empty circles

the fourth round of mutagenesis, where the highest degree of cleavage activity was detected (227-fold compared to wild-type, Fig. 1c), was deep sequenced revealing a large variety of substitutions throughout the nuclease sequence (Fig. 1d). As an intermediate step to verify the yeast evolution step, we sampled the cDNA library to evaluate genome editing improvement in mammalian cells. We selected the most frequent substitutions (E189G, F214I, E789K, T913S, Fig. 1d) which were tested individually for indel formation in three endogenous loci (*RHO*, *CCR5*, and *EMX*) of HEK293 cells. Each single mutation showed variable editing improvement (Fig. 1e), thus suggesting that enhancement in yeast corresponded to optimization in a mammalian context.

### Selection of the enhanced CjCas9 variants in a mammalian reporter cell line

To select the best combinations of mutations for CjCas9 enhancement in mammalian cells, we generated a reporter cell line carrying a switch-on EGFP circuit controlled by indel formation. HEK293 cells carrying a Tet repressor (TetR) cDNA were engineered with a EGFP cassette repressed by Tet operator (TetO) elements, which

are released by CjCas9 cleavage of TetR (Fig. 2a and Additional file 1: Fig. S2a). The Cas-mediated induction of fluorescence increases the dynamic range compared to the widely used EGFP disruption assay [19], thus facilitating the detection of differences in Cas activity (Additional file 1: Fig. S2a). A lentiviral library of the yeast-evolved CjCas9 variants was transduced in the TetR-EGFP reporter cell line to select the most active mutants (Additional file 1: Fig. S2b). The selection step was repeated a second time to further enrich the best variants, which showed almost tenfold higher fluorescence than controls, indicating the presence of mutants more active than the original enzyme (Fig. 2b and Additional file 1: Fig. S2b). A large number of variants ($n = 160$) were isolated from double-sorted EGFP cells and tested individually (Additional file 1: Fig. S2c). We selected the best-performing variants, corresponding to mutants having the highest ratio of positive EGFP cells with respect to CjCas9 WT in the TetR-EGFP reporter cell line. The best-performing variants ($n = 17$) were further evaluated for indel formation in the *EMX1*, *RHO*, and *CCR5* endogenous loci (Fig. 2c), revealing an average increased activity compared to wild-type ranging from 2- up to 3.6-fold (Additional file 2: Table S1). Interestingly, the activity fold change observed at the *EMX1* locus (mean 4.4-fold) is consistently larger compared to *RHO* and *CCR5* (mean 1.9-fold for both), possibly because the former locus is actively expressed in HEK293 cells, while the latter loci are not [22].

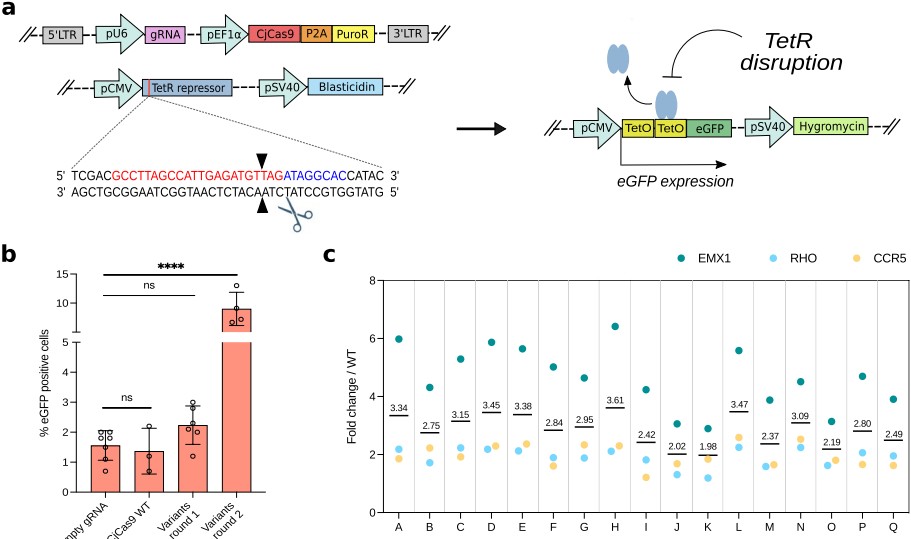

**Fig. 2** CjCas9 enhancement through screening in mammalian cells. **a** Schematic representation of the TetR-EGFP reporter cell line. TetR (blue) is constitutively expressed inactivating EGFP (green) expression. TetR inactivation by Cas9 cleavage enables EGFP expression and isolation of fluorescent cells by sorting. **b** CjCas9 variants isolated from rounds 1 and 2 of the mammalian screening are compared with CjCas9 WT transducing the TetR-EGFP reporter cell line. Statistical analysis was performed using one-way ANOVA with Tukey's multiple comparisons test: ns, not significant, ****$P < 0.0001$. **c** Editing activity of selected variants obtained from yeast and mammalian selection. Dots represent the fold changes of activity of each variant (from A to Q) with respect to CjCas9 WT targeting EMX, RHO, or CCR5. Extended data are shown in Additional file 1: Fig. S3. The grand mean values of each variant are shown in the graph. Sanger sequences of the variants are reported in Additional file 2: Table S2. In **b**, data are reported as mean ± standard deviation of $n \geq 3$ biologically independent samples. Individual values are represented as empty circles

In conclusion, through the evolution pressure obtained by coupling Cas cleavage activity with auxotrophic selection in yeast and indel formation in mammalian cells, EPICA enables the generation of enhanced Cas variants.

### UltraCjCas9, an evolved highly active Cas9

The most active variant, named UltraCjCas9 hereafter, was further characterized. Sanger sequencing revealed five modifications in four domains (RuvC-I, REC1, HNH, and PI) (Fig. 3a and Additional file 1: Fig. S4). Given the presence of one mutation in the PI domain (T913S), we verified the PAM preference of UltraCjCas9 through an in vitro PAM determination assay. The results showed that UltraCjCas9 recognizes a $N_4RYAY$ consensus sequence, relaxing positions 4 and 8 of the wild-type $N_3VRYAC$, thus expanding the targeting range (Fig. 3b and Additional file 1: Fig. S5). To evaluate the editing profile, we performed a comparative editing analysis in 23 human genomic loci with UltraCjCas9, CjCas9 WT, and the recently optimized enCjCas9 [15] showing that UltraCjCas9 was more active in the majority of sites (Fig. 3c). The editing efficacy of UltraCjCas9 was quite variable, reaching up to 12-fold higher activity than the wild-type (*HEK site 1*) (Fig. 3c and Additional file 2: Table S3). Overall, UltraCjCas9 was more active than CjCas9 WT (28.1% and 13.0% mean editing, respectively) and outperformed enCas9 (17.5% mean editing), which was obtained by rational engineering (Fig. 3d). Given the editing efficacy of UltraCjCas9, we compared this variant with the most commonly used Cas9 from *Streptococcus pyogenes*, SpCas9, by selecting genomic loci efficiently edited by UltraCjCas9 and using overlapping guides for both orthologs (Additional file 1: Fig. S6b). Overall, the two nucleases showed no significant difference in editing efficiency (Fig. 3e).

Finally, since enhanced Cas9 nuclease activity and relaxed PAM may correlate with decreased fidelity [23, 24], we performed a whole-genome off-target analysis by GUIDE-Seq [25] to compare the specificity of CjCas9 WT and UltraCjCas9. We chose two sites, *AAVS1-TS1* and *AAVS1-TS8*, previously associated with CjCas9 off-target activity [20], where both nucleases produced similar levels of on-target indels (Additional file 1: Fig. S6c). In both genomic loci, UltraCjCas9 generated less off-target cleavage sites than the wild-type enzyme, resulting in slightly higher precision (Fig. 3f). The similar precision profile was confirmed by comparing the sgRNA mismatch tolerance (single nucleotide mismatches) of CjCas9 WT and UltraCjCas9 (Additional file 1: Fig. S7). Overall, UltraCjCas9 exhibited a specificity profile similar to CjCas9 WT, confirming that the enhanced activity did not alter editing precision.

Overall, UltraCjCas9, generated by EPICA, showed an enhanced activity and an expanded targeting range while preserving specificity.

### Discussion

In just over a decade, CRISPR-Cas systems working as adaptive immunity in prokaryotes have been successfully translated into genome editing tools for mammalian cells up to the clinic. Notably, the prokaryotic cellular environment highly differs from the eukaryotic nuclear compartment, characterized by multiple factors defining chromatin structure and function. Hence, to further advance this technology, CRISPR-Cas systems should be functionally steered in eukaryotic cells. Along this line, emerging data from

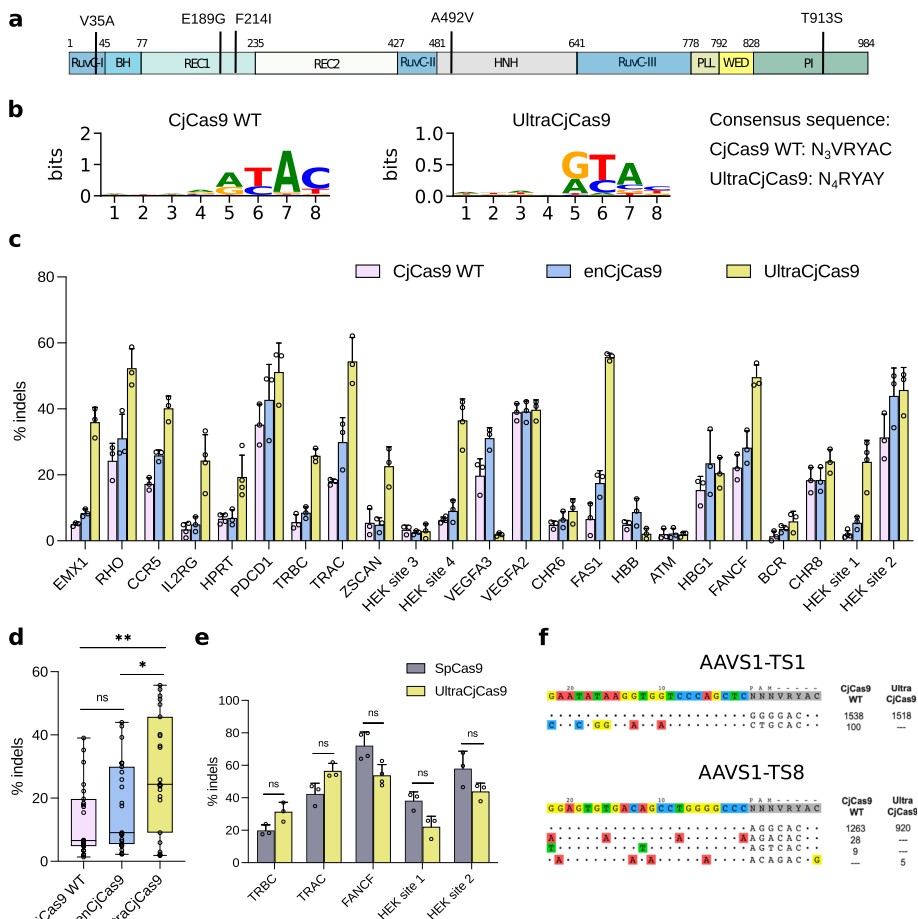

**Fig. 3** Characterization of the highly enhanced variant UltraCjCas9. **a** Domain localization of the five amino acid substitutions (V35A, E189G, F214I, A492V, and T913S) characterizing UltraCjCas9. See Additional file 1: Fig. S4 for amino acid positions in the CjCas9 structure. **b** Sequence logo representations of the PAM preference of CjCas9 and UltraCjCas9, obtained using an in vitro PAM determination assay. Heatmaps representing each nucleotide combination are reported in Additional file 1: Fig. S5. **c** Editing efficiency in the endogenous genomic loci of HEK293 cells with CjCas9 WT, enCjCas9, and UltraCjCas9. PAM sequences are reported in Additional file 1: Fig. S6a. **d** Graphical summary of editing activities of CjCas9 WT (pink), enCjCas9 (blue), and UltraCjCas9 (yellow) in **c**. Empty circles in the box plots represent the average percentages of indels for each genomic locus. Statistical analysis was performed using one-way ANOVA with Tukey's multiple comparisons test: ns, not significant, *$P < 0.05$, **$P < 0.01$. Central line, median; box limits, upper and lower quartiles; whiskers, × 1.5 interquartile range; $n = 23$ independent loci. **e** Comparison of SpCas9 and UltraCjCas9 editing activity in the indicated endogenous genomic loci. Statistical significance was assessed using a two-sided *t*-test corrected using the Holm-Šídák method for each locus. ns, not significant. **f** GUIDE-Seq analysis performed with CjCas9 WT and UltraCjCas9. GUIDE-seq read counts of each on- and off-target in HEK293T cells are shown on the right side (extended data in Additional file 2: Table S7–S10). In **c** and **e**, data are reported as mean ± standard deviation of $n \geq 3$ biologically independent samples. Individual values are represented as empty circles

various groups demonstrated that chromatin structure impacts CRISPR-Cas genome editing efficacy [26–34]. Remarkably, while nucleosomes affect the editing activity of CRISPR-Cas enzymes, the artificial zinc finger nucleases (ZFN) remain unaffected, suggesting a natural evolution of Cas nucleases to work with naked DNA [29]. More recently, direct evidence demonstrated the role of chromatin structure on Cas12a PAM recognition and R-loop formation efficiency [32] and the effect of chromatin remodeling

factors on the nuclease activity of a variety of Cas9 proteins, including small orthologs [33]. While the effect of chromatin structure on Cas efficiency has been well documented, other factors could influence nuclease activity in eukaryotic cells, such as interactions with host proteins. Post-translational modifications (PTMs) have been shown to inhibit Cas proteins in prokaryotes [34–36]; however, it remains unclear to what extent nuclease activity could be modulated by PTMs introduced by host factors in eukaryotes [37]. These results suggest that limited insights into Cas interactions with the eukaryotic environment will severely limit the molecular engineering of these tools.

While rational design has been widely employed to engineer Cas nucleases [15, 38–40], this approach requires protein structure information which is not necessarily available for the large number of RNA-guided nucleases requiring optimization as genome editing tools. Moreover, resolved protein structures are often missing flexible regions, such as the HNH domain of Cas9 nucleases [41–43], imposing limitations on the rational design approach. In general, enhancement of Cas9 nucleases by rational engineering is obtained through modifications of residues directly contacting the DNA-RNA hybrid, often increasing positive charges in order to facilitate interactions between the Cas9 protein and the negatively charged nucleic acid backbone [15, 39]. In line with this type of enhancing modifications, UltraCjCas9 carries the substitution E189G in the REC1 domain, which neutralizes a negative charge in the vicinity of the PAM-proximal region of the DNA-RNA hybrid. Conversely, the remaining mutations in UltraCjCas9 would have been hardly predictable through structural analysis, demonstrating the advantage of an unbiased in cellulo Cas9 selection over rational engineering [15]. In particular, mutations V35A and A492V are absent from the crystal structure of CjCas9 [41]. According to the AlphaFold prediction [44], V35A is located in a loop connecting the RuvC-I and BH domains, while A492V is located near the base of an alpha helix at the start of the HNH domain (Additional file 1: Fig. S4). Unexpectedly, these mutations are chemically similar to the wild-type residues, and their effect could be mediated by different steric hindrances of their side chains. Moreover, mutation F214I is located at the end of an alpha helix in the REC1 domain and is positioned far from DNA-RNA interactions in the crystal structure. Specifically, Phe214 is positioned near the cleft between REC1 and REC2, which accommodates the unmelted PAM-distal dsDNA during R-loop formation [45]. Finally, mutation T913S, located in the PAM interacting domain, is likely responsible for the PAM relaxation of UltraCjCas9. In CjCas9 WT, Thr913 forms a water-mediated hydrogen bond with the fourth PAM nucleotide and determines the specificity for V (A, C, or G) at this position due to the steric incompatibility of its side chain with the methyl group of thymine [41]. Mutation T913S likely avoids this steric clash thanks to the smaller side chain of serine compared to threonine. While structural considerations can explain the effect of this mutation on PAM preference, the structural basis for its effect on activity remains unclear. Interestingly, modifications introduced in UltraCjCas9 did not interfere with the nuclease specificity, as opposed to previous molecular engineering approaches [39], implying that enhanced nuclease activity does not necessarily increase tolerance to sgRNA-target DNA mismatches for Cas9 proteins, as also previously observed for CasΦ [46].

UltraCjCas9, obtained by directed evolution in EPICA, has attractive features for genome editing applications, including high editing efficiency (similar to SpCas9 at

specific genomic sites) and high specificity. Interestingly, while its editing activity is higher at most loci compared to CjCas9 WT, there are specific loci showing no increase in activity (e.g., *VEGFA2*, *HBG1*) or even a decrease (*VEGFA3*). This variability in editing activity is not uncommon for enhanced Cas nucleases [15, 47, 48] and could be ascribed to modified preferences for target sequence composition, which is known to affect Cas activity [49]. Finally, the reduced molecular size of UltraCjCas9 is compatible with the packaging capacity of AAV vectors, the most commonly used delivery method for genome editing [50, 51], consenting its delivery all-in-one with the sgRNA, as opposed to double AAVs for sgRNA and Cas9 for in vivo use [52–55].

## Conclusions

Here, we report a fully eukaryotic platform for Cas enhancement. Our methodology exploits evolutionary steps in yeast followed by a selection of variants in mammalian cells. Enhanced nucleases such as UltraCjCas9, obtained through EPICA, will foster the expansion of the genome editing toolbox with the development of a larger variety of CRISPR-Cas systems. Since the platform allows the evolution of enzymes with nuclease activity, its use can be potentially extended beyond CRISPR-Cas systems, including the novel emerging collection of RNA-guided nucleases associated with prokaryotic and eukaryotic transposons [3–5, 56]. The modification of Cas9 proteins steered under the pressure imposed by the complexity of the nuclear environment may shed light on critical residues required for nuclease enhancement in eukaryotes, thus facilitating the translation of prokaryotic resources for genome editing applications.

## Materials and methods

### Plasmids

The constructs used for the generation of the yeast strain were obtained by cloning the AAVS1-TS34 and AAVS1-TS32 targets and PAM sequences as annealed oligonucleotides in pUC19 plasmid containing respectively ADE and TRP cassettes, after digestion with KpnI and BamHI. For expression in yeast, the sequence of CjCas9 was amplified from the pX404 (Addgene 68338) and cloned in place of SpCas9 in the LEU2 carrying plasmid p415-GalL-Cas9-CYC1t (Addgene 43804), by double digestion with SpeI/XhoI. pSNR52-BsmBI-Cj_sgRNA for sgRNA expression in yeast was generated by substituting the promoter in the pU6 Cj sgRNA plasmid (Addgene 89753) with the SNR promoter by double digestion with XhoI and BamHI. The AAVS1-TS34 and AAVS1-TS32 target sequences were cloned as annealed oligonucleotides in the pSNR52 Cj sgRNA digested with BsmBI. The SNR − AAVS1-TS34 and − TS32 Cj sgRNA fragments were first cloned in the URA3 carrying pRS316 plasmid by double digestion with XhoI/SacII adding the SUP4t yeast terminator; for double guide plasmid the SNR−AAVS1TS32 Cj sgRNA fragment was then amplified and cloned in the generated pRS316-SNR52p-AAVS1TS-34gRNA − SUP4t plasmid by digestion with SacI. The Sp sgRNA plasmids were obtained through PCR site-directed mutagenesis of p426-SNR52p-gRNA.CAN1.Y-SUP4t (Addgene 43803) to introduce the target sequences. For mammalian expression, the CjCas9 sgRNA scaffold was optimized through site-directed mutagenesis to interrupt the A/T stretches [33]; targets were cloned as annealed oligos in the pU6 opt CjCas9 sgRNA cut with BsmBI. The list of sgRNAs cloning primers is available in Additional

file 2: Table S4. For expression in mammalian cells, CjCas9 WT was cloned in a pX330 containing the chicken β-actin promoter by double digestion with AgeI and EcoRI/BbsI. To generate single mutants, CjCas9 wild-type sequence was cloned in a pEGFP-N1 with AgeI and MfeI and single substitutions introduced through PCR site-directed mutagenesis; the protein sequence was then cloned back in pX330. Multiple mutants were amplified from the two rounds enriched sorted cells and cloned in pX330 downstream of a SV40 NLS sequence previously inserted in the plasmid, as well as CjCas9 WT, digesting with KpnI and EcoRI/BbsI. enCjCas9 was generated as a double mutant introducing L58Y and D900K mutations in the CjCas9 WT sequence through site-directed mutagenesis and then cloned in the pX330 NLS plasmid. Primers used for plasmid cloning are reported in Additional file 2: Table S5.

## Yeast culture

Yeast strains were grown in a YPDA-rich medium; for auxotrophic selection, yeasts were grown in a synthetic minimal medium (SD), omitting the single amino acids required for the experimental purpose. Yeasts stably expressing CjCas9 sgRNA were kept growing in SD without uracil before transformation. Cas9 expression was obtained using 20 g/L D-(+)-galactose and 10 g/L D-(+)-raffinose instead of dextrose.

## Cell culture

HEK293 and HEK293T from the American Type Culture Collection (ATCC) were cultured in Dulbecco's modified Eagle's medium (DMEM; Life Technologies) with 10% FCS (Life Technologies) and antibiotics (Life Technologies). The EGFP T-REx 293 cell line used for the mammalian screening was obtained by modifying the Flp-In™ T-REx™ 293 Cell Line (Invitrogen) for the expression of inducible EGFP according to the manufacturer's protocol. The cell lines were verified for the absence of mycoplasma contamination (PlasmoTest, Invitrogen).

## Yeast screening

The yeast strain used for the screening was generated from the yLFM-ICORE strain by the Delitto Perfetto approach [57], using the constructs pUC19-Ade2-AAVS1TS34 and pUC19-Trp1-AAVS1TS32. The library of CjCas9 variants was produced by random mutagenesis of the wild-type sequence using error-prone PCR approach (GeneMorph II kit from Agilent) (for primers, see Additional file 2: Table S5). The manufacturer's conditions were followed to set a rate of 0–4.5 mutations/kb; yeasts were co-transformed with the PCR library and the previously digested (SpeI/XhoI) p415-GalL-CYC1t plasmid in a 3:1 ratio, to obtain direct assembly in vivo. Yeasts were transformed following the Gietz et al. published protocol [58]. After transformation, yeasts were grown for 5 h in an SD medium lacking uracil and leucine for recovery and recombination, then the expression of Cas9 was induced through overnight incubation in a galactose-containing medium. The next day, yeasts were plated in a medium lacking adenine and tryptophan, to select the colonies with enhanced variants. Colonies were collected, and the plasmids were extracted with Zymoprep Yeast Plasmid Miniprep II (Zymo Research). For the next rounds of mutagenesis, high-fidelity PCR was performed on the selected variants with Phusion™ High-Fidelity DNA Polymerase (Thermo Scientific) and used as a template for

the subsequent step of mutagenesis (described above). For each round of mutagenesis, transformation and collection of selected colonies were repeated. For the validation of the yeast strain, cells were transformed with the entire p415 plasmid expressing wild-type SpCas9 or CjCas9, using the same conditions of the screening. Yeast colonies were counted using the ImageJ software, manually setting the threshold for the maximum discrimination of the colonies. Each yeast experiment was performed by plating a defined amount of yeast suspension in $n = 3$ selective plates as technical replicates.

### Deep sequencing

Variants of the fourth round of mutagenesis were PCR amplified with Phusion High-Fidelity DNA Polymerase (for primers, see Additional file 2: Table S5); the amplicon was used to generate a library with Nextera DNA Flex Library Preparation Kit (Illumina) and sequenced in a PE150 run using Illumina Miseq Reagent kit V2. For data analysis, reads were trimmed using TrimGalore [59] version 0.6.6 (with parameters -q 30 –paired) and aligned to the CjCas9 coding sequence using bowtie2 [60] version 2.3.4.3. Pileups were generated using PaCBAM [61] version 1.6.0 and a custom Python script was used to analyze mutation frequencies.

### Indels analysis at human genomic loci

For the evaluation of genome editing activity, $10^5$ HEK293 cells were seeded in 24-well plates and transfected the next day with 200 ng of sgRNA plasmid and 400 ng of Cas9 plasmid; TransIT-LT1 (Mirus Bio) was used as transfection reagent according to manufacturer's protocol. Cells were collected after 3 days and lysed with QuickExtract DNA Extraction Solution (Lucigen): endogenous loci were PCR amplified with HOT FIREPol MultiPlex Mix (Solis BioDyne), Sanger sequenced and analyzed with the TIDE tool [62]. The list of TIDE analysis primers is available in Additional file 2: Table S6.

### Lentiviral vector library construction

The CjCas9 sgRNA targeting the TetR repressor was amplified from the pU6 plasmid and cloned in the LentiCRISPR V1 by double digestion with NdeI and EcoRI. LentiCRISPR V1 sgRNA TetR was then modified to have an intron upstream of the cloning site of Cas9, to avoid a leaky expression of the CjCas9 variants in bacteria and the potential loss of clones. An intron previously used for the same purpose [63] was cloned by insertional mutagenesis through PCR in the middle of the CjCas9 NLS sequence, between a splicing donor site and a splicing acceptor site: the construct Kozak-NLS-V5-intron was then amplified adding a KpnI site downstream and was cloned in LentiCRISPR using the restriction enzymes BamHI and NheI. To minimize the number of background colonies with undigested plasmid, the bacterial toxin ccdB was cloned with the enzymes KpnI and NheI using a dedicated commercial ccdB-resistant strain (Thermo Fisher); the generated plasmid LentiCRISPRV1-gRNA-TetR-intron-ccdB was used to obtain the KpnI/NheI digested backbone for lentiviral library. CjCas9 variants were amplified from the plasmid extract of the fourth round of the yeast screening, digested with KpnI and NheI and ligated at room temperature for 2 h with the digested backbone. The ligation was purified by NucleoSpin Gel and PCR Clean-up (Macherey–Nagel) and used for the electroporation of ElectroMAX

DH5α-E Competent Cells (Invitrogen): bacterial colonies were collected until a coverage of at least $100 \times$ was reached. The number of different variants was estimated to be 1398, based on the total number of colonies collected from the fourth round of the yeast screening. Pooled library plasmids were purified using the NucleoBond PC 500 Maxi kit (Macherey–Nagel). The same protocol was applied to generate the library for the second round of selection, amplifying the variants from the genomic DNA of first-round sorted cells. Primers for the amplification were lengthened for each library to avoid the carry-over of unspecific products from the previous cloning. Primers used for lentiviral vector cloning are reported in Additional file 2: Table S5.

### Cell transduction and sorting

For the lentiviral library production, $2 \times 10^7$ HEK293T cells (American Type Culture Collection) were transfected with 25 μg of LentiCRISPR sgRNA TetR intron CjCas9 library, 16.2 μg of pCMV-deltaR8.91 and 8.7 μg of VSV-G using the polyethylenimine (PEI) method. Lentiviral particles were filtered through a 0.45-μm PES filter and concentrated with ultracentrifugation. Vectors were resuspended in Opti-MEM (Gibco) and conserved at $-80$ °C. MOI was estimated by transducing a defined cell number with growing volumes of lentiviral particles and evaluating their viability in puromycin (1 μg/m) after 2 days using an MTT assay. For the rounds of sorting, EGFP T-REx 293 cells were seeded and the next day were transduced with the vector library at MOI $\sim 0.3$, ensuring a $> 100 \times$ coverage of the library variants. Cells were then selected in puromycin (1 μg/ml) 2 days after transduction and sorted for their EGFP fluorescence after 6 additional days using FACS ARIA III (BD Biosciences). Cells were grown for recovery and collected for genomic DNA extraction with a DNeasy Blood and Tissue kit (Qiagen).

### In vitro *PAM determination assay*

The PAM identification assay was performed through in vitro cleavage of a randomized PAM library, adapting a previously established protocol (Altae-Tran et al. 2021). In brief, CjCas9 and UltraCjCas9 were in vitro transcribed and translated using PURExpress In vitro protein synthesis kit (NEB), along with their sgRNA targeting the PAM library spacer. Primers used are reported in Additional file 2: Table S5. In vitro cleavage was performed at the same time, adding 25 ng of randomized PAM library, previously created in our lab, in the mix and incubating the reaction for 4 h at 37°. The reaction was stopped by placing it on ice for 10 min and adding RNase A/T1 (Thermo Fisher) and Proteinase K (Thermo Fisher) to remove RNA and proteins. DNA was purified using $0.7 \times$ NGSclean beads (CleanNA): the ends were filled-in using T4 DNA polymerase (Thermo Fisher) and dA-tailed using DreamTaq DNA polymerase (Thermo Fisher). dsDNA oligo adapters were ligated to the fragments and a first PCR was performed as previously described [9]. The fragments of 200 bp were recovered from a size-selection purification with 0.5X NGSclean beads (CleanNA), purified with 1X NGSclean beads, and used as a template for a second round of PCR to attach Illumina indexes (NEBNext Multiplex Oligos for Illumina). Deep sequencing and data analysis were performed as previously described [9].

### GUIDE-Seq

GUIDE-Seq was performed following a previously published protocol with minor modifications [25]. HEK293T cells (ATCC) were transfected with 250 ng of sgRNA plasmid, 500 ng of Cas9 plasmid, and 10 pmol of dsODNs, using Lipofectamine 3000 (Invitrogen). Cells were selected in puromycin (1 µg/ml) the next day, and after 4 days, genomic DNA was extracted using a DNeasy Blood and Tissue kit (Qiagen). Genomic DNA was sonicated to an average size of 500 bp using an S2 Focused-ultrasonicator (Covaris) and end-repaired with NEBNext Ultra End Repair/dA Tailing Module. Adaptors were ligated using the NEBNext Ultra Ligation Module as previously described [64]. After PCR amplifications, libraries were quantified by Qubit dsDNA High Sensitivity Assay kit (Invitrogen) and sequenced with the MiSeq sequencing system (Illumina) using Miseq Reagent kit V2-150PE.

### CjCas9 structure

The crystal structure of CjCas9 WT in complex with its sgRNA and target DNA was retrieved from PDB (accession 5X2H) [41]. To investigate mutations in missing regions, the AlphaFold [65] predicted structure was retrieved from the AlphaFold Protein Structure Database (accession Q0P897) [44] and was aligned to the crystal structure using the Dali server [66]. Protein structures were visualized using ChimeraX [67].

### Statistical analysis

Statistical significance tests were performed using GraphPad Prism (version 9.4.1). Data in Figs. 2b and 3d were analyzed with a one-way ANOVA followed by Tukey's multiple comparisons test. Data in Fig. 3e were analyzed using two-sided *t*-tests corrected for multiple comparisons using the Holm-Šídák method. For all analyses, adjusted *p*-values less than 0.05 were considered statistically significant.

### Supplementary Information

---

**Additional file 1: Fig. S1.** Yeast cultures from the auxotrophic evolution platform. **Fig. S2.** TetR-EGFP reporter screening platform. **Fig. S3.** Editing activity of selected variants in endogenous loci of HEK293 cells. **Fig. S4.** 3D position of mutations in UltraCjCas9. **Fig. S5.** In vitro PAM characterization of CjCas9 WT and UltraCjCas9. **Fig. S6.** List of PAM sequences and editing at GUIDE-Seq ON-target sites. Fig. S7. Editing activity of CjCas9 WT and UltraCjCas9 in the *PDCD1* locus using mismatched sgRNAs.

**Additional file. 2: Table S1.** Indel frequencies of selected variants. **Table S2.** Sanger sequences of selected variants. **Table S3.** Indel frequencies on genomic loci. **Table S4.** sgRNA cloning primers. **Table S5.** Cloning primers. **Table S6.** TIDE analysis primers. **Table S7.** Identified off-targets of CjCas9 WT targeting AAVS1TS1. **Table S8.** Identified off-targets of UltraCjCas9 targeting AAVS1TS1. **Table S9.** Identified off-targets of CjCas9 WT targeting AAVS1TS8. **Table S10.** Identified off-targets of UltraCjCas9 targeting AAVS1TS8. **Table S11.** Indel frequencies with mismatched sgRNAs.

**Additional file 3.** Review history.

---

### Acknowledgements

We are grateful to Cereseto's lab for the helpful discussion throughout the project. We thank the next-generation sequencing facility at the University of Trento for the technical support.

### Peer review information

**Review history**

The review history is available as Additional file 3.

**Authors' contributions**

G.V.R., E.K., M.D.G., and S.A. designed and performed the experiments. M.C. performed the computational analyses and analyzed the data. G.V.R., M.C., A.Ca., and A.Ce. wrote and edited the paper. A.Ce. and A.Ca. conceived and designed the study. A.Ce. was responsible for the coordination of the study. All authors read, corrected, and approved the final manuscript.

**Funding**

This work was supported by the European Union's Horizon 2020 innovation program through the UPGRADE (Unlocking Precision Gene Therapy) project (grant agreement No 825825) and Horizon Europe EIC Pathfinder Program AAVolution (grant agreement 01071041).

**Availability of data and materials**

Deep sequencing data of the CjCas9 library derived from the 4th round of mutagenesis and GUIDE-Seq data for CjCast WT and UltraCjCas9 are available on NCBI (BioProject PRJNA1020659) [68]. The custom code used in this study is available on Zenodo and GitHub: https://github.com/Matteo-Ciciani/EPICA [69, 70].

## Declarations

**Ethics approval and consent to participate.**

Not applicable.

**Consent for publication**

Not applicable.

**Competing interests**

The authors declare competing financial interests: A.Ce. is a co-founder and holds shares of Alia Therapeutics, a genome editing company. A.Ca. is a co-founder, holds shares, and is currently an employee of Alia Therapeutics. M.C. is a consultant of Alia Therapeutics.

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

## 