## [**Additional file 3.** Review history. · Genome Biology]

Review History

First round of review

Reviewer 1

Are you able to assess all statistics in the manuscript, including the appropriateness of statistical tests used? Yes, and I have assessed the statistics in my report.

Comments to author:

Ruta et al present a methodology to perform activity optimization of crisper-cas enzymes in Eukaryotic systems. Methodologies on that direction are very much needed given the fact the large number of discovered cas enzymes which are not highly active in eukaryotic system. Although cas activity is somewhat portable prokaryote -> eukaryote, numerous important context differences exist including chromatin landscape and much larger genome size. It is very worth for the field to have validated eukaryotic evolution systems. Also improved cjcas9 (which is smaller than most used cas9 enzymes) are provided.

- Are the methods appropriate to the aims of the study, are they well described, and are necessary controls included? If not, please specify what is required.

The methodology is overall appropriate. I only have two comments below:

The authors select a 4 mutations to analyze in detail. It seems clear a positive trend with the new mutations and promising portability. One way to confirm the results and expand the dynamic range may to combine the mutations.

In figure 3b, activity in a large number of sites is reported. A minority of sites show little improvement (or even a negative trend - ie VEGFA3). Could the authors comment on this? Any concerns on PAM specificity alteration?

- Are the conclusions adequately supported by the data shown? If not, please explain

I highlighted a few points which I believe require further justification/study (see below):

from figures 1 and 2, it seems that the de novo mutations benefit more the EMX1 site over the others. This issue should be mentioned and analyzed in the discussion section

UltraCjCas9 is presented in a diagram on fig 3 which is nice. However, description of the mutations and potential interpretation is missing from the discussion section. Although it is mentioned that they could be hardly been predicted from rational design, It would be worth to provide some discussion on potential interpretations using an structural model.

- Are sufficient details provided to allow replication and comparison with related analyses that may have been performed? If not, please specify what is required.

generally methodologies are well described

- Does the work represent a significant advance over previously published studies?

yes. other studies reported improved cjCas9 (ie Nakagawa et al, Comm bio 2022; cited in the study) but this manuscript reports 1) a much improved cjas9 enzyme is provided 2) a generalizable framework for cas evolution is described

- Is the paper of broad interest to others in the field, or of outstanding interest to a broad audience of biologists?

Cas evolution is a very timely topic in gene editing. Providing validated methods for evolution in eukaryotic and mammalian context it is very valuable.

Reviewer 2

Are you able to assess all statistics in the manuscript, including the appropriateness of statistical tests used? Yes, and I have assessed the statistics in my report.

Comments to author:

In this article, the author employed CjCas9 as the candidate for engineering and conducted directed evolution using a eukaryotic cell platform comprised of yeast and HEK293 cells. The mutant libraries were constructed through error-prone PCR in vitro. For yeast screening, target loci were inserted into the coding sequences of the TRP1 and ADE2 genes within the yeast genome. Inserting DNA segments between these genes caused frameshift mutations that prevented the yeast from growing in media lacking tryptophan or adenine. The authors took advantage of the strong homologous recombination activity in yeast, which allowed the conversion of TRP1 and ADE2 to wildtype at indel sites via homologous recombination. This enabled the yeast cells to survive in media without tryptophan and adenine. The authors carried out four iterative rounds of discontinuous selection, in which the CjCas9 from amplified through error-prone PCR from survivors were amplified for subsequent rounds of error-prone PCR, creating new mutant libraries for further screening in yeast. This process was repeated for four rounds. To avoid bias, the researcher employed these two different target sites. This approach is akin to a previously documented strategy by the same corresponding author, which yielded the high-fidelity evoCas9 (refer to PMID: 29431739, published in Nat Biotechnol., 2018).

Following the yeast screening, the mutant library was cloned into a lentiviral vector and introduced into HEK293 cells, which had been engineered to contain the Tet repressor (TetR) and an eGFP reporter gene regulated by a tet operator (TetR-eGFP reporter). Cleavage of TetR by CjCas9 enabled normal expression of eGFP. GFP-positive cells were isolated using flow cytometry, and the CjCas9 mutants from these cells were recloned into the lentiviral vector for additional testing in TetR-eGFP reporter cells. The most active mutant was named UltraCjCas9. The author found that disrupting TetR provided a more sensitive screening assay than disrupting eGFP.

When compared with the WT-CjCas9 and the rational-engineered version enCjCas9 (PMID: 35260779), UltraCjCas9 showed much higher indel activity, with up to 12 times the

effectiveness of the original enzyme. GUIDE-seq results showed that UltraCjCas9 displayed reduced off-targeting activity while retained comparable on-target efficiency.

Overall, this reviewer felt the experiments were well-organized, and the evolutionary strategy was convincing and robust, and the data were well presented. The authors pushed the genome editing field forward by providing a thoroughly tested UltraCjCas9 variant, which exhibited enhanced genomic editing capabilities. This reviewer only has a few minor points for the authors to address.

1. In Supplementary Figure 2a, the author showed that the TetR disruption was a more sensitive assay than directly disrupting the eGFP. Can the authors elaborate why this was the case. The Flp-In™ T-REx™ 293 Cell Line used in the experiment did not specify the copy number of the TetR gene. If the cell line has multiple TetR copies, Cas9 has to disrupt all copies to survive the screen. Could this be the reason behind heightened sensitivity?

2. In Figure 3e, the author tested the specificity of UltraCjCas9 at two sites using GUIDE-seq. An additional experiment assessing gRNA mismatch tolerance (refer to PMID: 26628643, Figure 3) would be desirable to further elucidate the specificity of UltraCjCas9.

Point-by-point responses to Reviewers' comments

Reviewer #1: Ruta et al present a methodology to perform activity optimization of crisper-cas enzymes in Eukaryotic systems. Methodologies on that direction are very much needed given the fact the large number of discovered cas enzymes which are not highly active in eukaryotic system. Although cas activity is somewhat portable prokaryote -> eukaryote, numerous important context differences exist including chromatin landscape and much larger genome size. It is very worth for the field to have validated eukaryotic evolution systems. Also improved cjcas9 (which is smaller than most used cas9 enzymes) are provided.

Thank you for the encouraging comments. We share your enthusiasm and emphasize that it is important for the broader community to collectively contribute to the advancement of genome editing technology with new experimental set up and novel Cas9 variants.

- Are the methods appropriate to the aims of the study, are they well described, and are necessary controls included? If not, please specify what is required.

The methodology is overall appropriate. I only have two comments below:

The authors select a 4 mutations to analyze in detail. It seems clear a positive trend with the new mutations and promising portability. One way to confirm the results and expand the dynamic range may to combine the mutations.

Discussion: The 4 most common mutations that emerged from the directed evolution in yeast were tested individually to verify that the yeast evolution was producing mutations enhancing the Cas variants. Since the evolution pipeline is composed of yeast and mammalian selection rounds, we considered relevant monitoring the enhancement trend rather than deepening the analysis of variants that had to be further selected in mammalian cells.

In figure 3b, activity in a large number of sites is reported. A minority of sites show little improvement (or even a negative trend - ie VEGFA3). Could the authors comment on this? Any concerns on PAM specificity alteration?

Discussion: Thank you for this comment. Indeed as shown in Figure 3b, the editing profile is quite heterogeneous: in specific sites the editing improvement with UltraCjCas9 is stronger than in others. This variability in editing efficiency at different loci is not uncommon for engineered Cas nucleases with improved activity (e.g. Xu et al. Mol Cell 2021, Nakagawa et al. Commun Biol 2022, Hino et al. Cell 2023). We added these considerations to the discussion.

Action: Stimulated by the question about PAM specificity we experimentally verified UltraCjCas9 PAM preference through a PAM assay. The PAM assay results (now reported in Figure 3b and Supplementary Figure 5) indeed show that the PAM of UltraCjCas9 is modified compared to wild-type, resulting in a consensus N₄RYAY PAM. Interestingly, this is a more relaxed sequence compared to the WT N₃VRYAC PAM and is likely the result of the T913S substitution in the PAM interacting domain. Even though the PAM relaxation cannot explain the diverse editing efficacy compared to WT CjCas9 (since UltraCjCas9 clearly also recognizes the WT PAM) the PAM relaxation provides a further advantage by extending the targeting range of this new variant.

We truly thank you for raising the PAM concern which allowed us to describe a further beneficial property of UltraCjCas9.

- Are the conclusions adequately supported by the data shown? If not, please explain I highlighted a few points which I believe require further justification/study (see below): from figures 1 and 2, it seems that the de novo mutations benefit more the EMX1 site over the others. This issue should be mentioned and analyzed in the discussion section

Discussion: As mentioned above, the variability in Cas enhancement in different loci was reported before with engineered Cas nucleases (e.g. Xu et al. Mol Cell 2021, Nakagawa et al. Commun Biol 2022, Hino et al. Cell 2023). We can speculate that this could depend on chromatin heterogeneity which may have different impacts on variants with diverse enzymatic kinetics. E.g. the EMX1 locus is actively transcribed in HEK293 cell

(<https://www.proteinatlas.org/ENSG00000135638-EMX1/cell+line>), while the CCR5 and RHO loci aren't (<https://www.proteinatlas.org/ENSG00000163914-RHO/cell+line> and

<https://www.proteinatlas.org/ENSG00000160791-CCR5/cell+line>). The open chromatin state of EMX1 may be responsible for the large boost in editing efficiency of enhanced variants compared to the other tested loci.

Action: we have revised the manuscript (results section 'Selection of the enhanced CjCas9 variants in a mammalian reporter cell line') to bring more specific attention to the editing variability and provided a possible explanation by citing former studies.

UltraCjCas9 is presented in a diagram on fig 3 which is nice. However, description of the mutations and potential interpretation is missing from the discussion section. Although it is mentioned that they could be hardly been predicted from rational design, It would be worth to provide some discussion on potential interpretations using an structural model.

Discussion: Thank you for this suggestion.

Action: We have expanded the discussion with structural considerations on the effects of the mutations.

- Are sufficient details provided to allow replication and comparison with related analyses that may have been performed? If not, please specify what is required.
generally methodologies are well described

Thank you for acknowledging proper description of the methodologies.

- Does the work represent a significant advance over previously published studies?
yes. other studies reported improved cjCas9 (ie Nakagawa et al, Comm bio 2022; cited in the study) but this manuscript reports 1) a much improved cjcas9 enzyme is provided 2) a generalizable framework for cas evolution is described

Thank you for acknowledging the contribution of our work for the advancement of genome editing technologies.

- Is the paper of broad interest to others in the field, or of outstanding interest to a broad audience of biologists?
Cas evolution is a very timely topic in gene editing. Providing validated methods for evolution in eukaryotic and mammalian context it is very valuable.

Thank you for positive and constructive feedback.

Reviewer #2: In this article, the author employed CjCas9 as the candidate for engineering and conducted directed evolution using a eukaryotic cell platform comprised of yeast and HEK293 cells. The mutant libraries were constructed through error-prone PCR in vitro. For yeast screening, target loci were inserted into the coding sequences of the TRP1 and ADE2 genes within the yeast genome. Inserting DNA segments between these genes caused frameshift mutations that prevented the yeast from growing in media lacking tryptophan or adenine. The authors took advantage of the strong homologous recombination activity in yeast, which allowed the conversion of TRP1 and ADE2 to wildtype at indel sites via homologous recombination. This enabled the yeast cells to survive in media without tryptophan and adenine. The authors carried out four iterative rounds of discontinuous selection, in which the CjCas9 from amplified through error-prone PCR from survivors were amplified for subsequent

rounds of error-prone PCR, creating new mutant libraries for further screening in yeast. This process was repeated for four rounds. To avoid bias, the researcher employed these two different target sites. This approach is akin to a previously documented strategy by the same corresponding author, which yielded the high-fidelity evoCas9 (refer to PMID: 29431739, published in Nat Biotechnol., 2018).

Following the yeast screening, the mutant library was cloned into a lentiviral vector and introduced into HEK293 cells, which had been engineered to contain the Tet repressor (TetR) and an eGFP reporter gene regulated by a tet operator (TetR-eGFP reporter). Cleavage of TetR by CjCas9 enabled normal expression of eGFP. GFP-positive cells were isolated using flow cytometry, and the CjCas9 mutants from these cells were recloned into the lentiviral vector for additional testing in TetR-eGFP reporter cells. The most active mutant was named UltraCjCas9. The author found that disrupting TetR provided a more sensitive screening assay than disrupting eGFP.

When compared with the WT-CjCas9 and the rational-engineered version enCjCas9 (PMID: 35260779), UltraCjCas9 showed much higher indel activity, with up to 12 times the effectiveness of the original enzyme. GUIDE-seq results showed that UltraCjCas9 displayed reduced off-targeting activity while retained comparable on-target efficiency.

Overall, this reviewer felt the experiments were well-organized, and the evolutionary strategy was convincing and robust, and the data were well presented. The authors pushed the genome editing field forward by providing a thoroughly tested UltraCjCas9 variant, which exhibited enhanced genomic editing capabilities. This reviewer only has a few minor points for the authors to address.

Thank you very much for the encouraging comments!

1. In Supplementary Figure 2a, the author showed that the TetR disruption was a more sensitive assay than directly disrupting the eGFP. Can the authors elaborate why this was the case. The Flp-In™ T-REx™ 293 Cell Line used in the experiment did not specify the copy number of the TetR gene. If the cell line has multiple TetR copies, Cas9 has to disrupt all copies to survive the screen. Could this be the reason behind heightened sensitivity?

Discussion: Thank you for your question. Rather than increased “sensitivity” the TetR is more “stringent with convenient dynamic range” to detect improved CjCas9 variants. With both reporter systems the activity of the control SpCas9 was similar (around 40%) showing no differential sensitivity, therefore the term “sensitivity” was not properly used. The advantage of TetR is in its stringency, with CjCas9 WT showing very low activity (close to background), while its activity is close to that of SpCas9 in the EGFP disruption assay (around 30%). This leads to a wide dynamic range, facilitating the detection of variants with improved activity with respect to the original CjCas9 WT.

Action: We have clarified this in the revised manuscript (results section 'Selection of the enhanced CjCas9 variants in a mammalian reporter cell line') and in the legend of Supplementary Figure 2a.

2. In Figure 3e, the author tested the specificity of UltraCjCas9 at two sites using GUIDE-seq. An additional experiment assessing sgRNA mismatch tolerance (refer to PMID: 26628643, Figure 3) would be desirable to further elucidate the specificity of UltraCjCas9.

Discussion: Thank you for this suggestion. We agree with the reviewer that, given the importance of preserved precision, a more thorough off-target analysis will improve the robustness of this work.

Action: We performed the suggested experiment, testing sgRNA mismatch tolerance of CjCas9 and UltraCjCas9 at the *PDCD1* locus using guides carrying single mismatches. Supplementary Figure 7 shows that, overall, UltraCjCas9 exhibits similar specificity to the WT variant, confirming that the increase in activity did not compromise its precision.

Second round of review

Reviewer 1

I have no further comment. Congratulations for this work.